# The Apoptosis Inhibitor Protein Survivin Is a Critical Cytoprotective Resistor against Silica-Based Nanotoxicity

**DOI:** 10.3390/nano13182546

**Published:** 2023-09-12

**Authors:** Christina Breder-Bonk, Dominic Docter, Matthias Barz, Sebastian Strieth, Shirley K. Knauer, Désirée Gül, Roland H. Stauber

**Affiliations:** 1Molecular and Cellular Oncology, University Medical Center Mainz, Langenbeckstrasse 1, 55101 Mainz, Germany; docter@uni-mainz.de (D.D.); rstauber@uni-mainz.de (R.H.S.); 2Leiden Academic Center for Drug Research (LACDR), Leiden University, Einsteinweg 55, 2333 CC Leiden, The Netherlands; barz@uni-mainz.de; 3Department of Dermatology, University Medical Center of the Johannes Gutenberg University Mainz, Langenbeckstraße 1, 55131 Mainz, Germany; 4Department of Otorhinolaryngology, University Medical Center Bonn, Venusberg-Campus 1, 53127 Bonn, Germany; sebastian.strieth@ukbonn.de; 5Center for Medical Biotechnology (ZMB), Department of Molecular Biology II, University of Duisburg-Essen, Universitätsstrasse 5, 45141 Essen, Germany; shirley.knauer@uni-due.de

**Keywords:** alveolar-capillary barrier, lung model, nanotoxicity, amorphous silica nanoparticles, inflammation, cytotoxic response

## Abstract

Exposure to nanoparticles is inevitable as they become widely used in industry, cosmetics, and foods. However, knowledge of their (patho)physiological effects on biological entry routes of the human body and their underlying molecular mechanisms is still fragmented. Here, we examined the molecular effects of amorphous silica nanoparticles (aSiNPs) on cell lines mimicking the alveolar-capillary barrier of the lung. After state-of-the-art characterization of the used aSiNPs and the cell model, we performed cell viability-based assays and a protein analysis to determine the aSiNP-induced cell toxicity and underlying signaling mechanisms. We revealed that aSiNPs induce apoptosis in a dose-, time-, and size-dependent manner. aSiNP-induced toxicity involves the inhibition of pro-survival pathways, such as PI3K/AKT and ERK signaling, correlating with reduced expression of the anti-apoptotic protein Survivin on the protein and transcriptional levels. Furthermore, induced Survivin overexpression mediated resistance against aSiNP-toxicity. Thus, we present the first experimental evidence suggesting Survivin as a critical cytoprotective resistor against silica-based nanotoxicity, which may also play a role in responses to other NPs. Although Survivin’s relevance as a biomarker for nanotoxicity needs to be demonstrated in vivo, our data give general impetus to investigate the pharmacological modulation of Survivin`s functions to attenuate the harmful effects of acute or chronic inhalative NP exposure.

## 1. Introduction

Over the past 20 years, nanomaterials have gained tremendous importance for industry. Due to multiple beneficial properties, the produced number of nanoparticles (NPs) and NP-containing products in the cosmetic and food industries is rising every year [1]. This development has led to an increased occurrence of NPs and thus NP exposure to the human body. However, the detailed effects of NP at nano-biological interfaces are still fragmented. NPs can enter the body by passing through biological diffusion barriers such as the skin (epidermis), the gastrointestinal tract (gastrodermis), or the alveolar-capillary barrier of the lung (respiratory epithelium). These cell-based barriers serve two opposing aspects by distinguishing different particles, molecules, and substances for either passage (i.e., gas exchange at the alveolar-capillary barrier) or rejection (hazardous substances). The alveolar region lines an area of 100–140 m^2^ of the lung and is therefore an attractive target for drug and gene delivery but also represents a critical point for harmful nanosized particles. Regarding the increasing use of drugs and drug carriers in the form of nanoparticles, it is of great importance to evaluate their interaction profile with biological barriers to determine possible treatment problems and undesired toxicity effects.

Synthetic non-metal silica (SiO_2_) nanoparticles (SiNPs) rank with 1.5 million tons among the most synthesized NPs used in hundreds of products, such as photovoltaic, tire compounds, or electrical and thermal insulation materials [2,3]. Furthermore, daily-use products such as cosmetics or toothpaste contain SiNP [4,5]. SiNP can be differentiated into crystalline and amorphous SiNPs exhibiting different chemical as well as biological properties. Crystalline SiNPs (crSiNP) possess a high reactivity responsible for several lung diseases like emphysema, silicosis, cancer, or pulmonary tuberculosis [6]. In contrast to crSiNP, the amorphous form of SiNP (aSiNP) has been considered to exhibit low toxicity [7,8]. Furthermore, the main benefits of aSiNP use combine an uncomplicated cost-saving production with the possibility for manifold surface functionalizations [9,10]. The synthesis of aSiNP can be adjusted for higher biodegradability [11]. For this reason, aSiNPs have quickly been applied in the industrial, medicinal, and other technological fields, and are a Food and Drug Administration (FDA) approved food additive [12].

However, despite their FDA approval, it has been shown that aSiNP can cause serious health implications. Several in vitro studies on nanoscale aSiNPs indicate cytotoxic and inflammatory effects and suggest that the NPs are taken up by digestion, skin absorption, and especially by inhalation [13,14]. Several types of NPs including aSiNPs possess the ability to translocate rapidly from the alveolar surface into the body [15]. Beyond these unintentional uptake routes, the active application of aSiNP as a drug or drug carrier system leads to the tremendously increasing presence of aSiNP in the blood system. The enhanced presence of aSiNP in the bloodstream is followed by a fast accumulation of aSiNP in tissue regions. The accumulation of aSiNP can cause dose-dependent cytotoxic effects and, in turn, can lead to organ injuries or at worst to organ failures [16,17].

In previous studies, it has been suggested that aSiNP can lead to cell death by inducing apoptosis in various cell lines [18,19,20]. The cytotoxicity of NPs could result from nanoparticle-mediated enzyme inhibition, and reactive oxygen species (ROS) production resulting in DNA and lipid modifications, but those effects are impaired by the formation of a protein corona enclosing the nanoparticle surface. However, in general, apoptosis is positively regulated by intrinsic receptor–ligand-mediated signaling processes or via mitochondrial-generated stimuli. The apoptotic cascade is negatively regulated by several pro-survival pathways that prevent the initiation of cell death [21]. One molecular actor involved in these pathways is a member of the inhibitor of apoptosis protein (IAP) family, Survivin/BIRC5 [22]. In normal cells, the Survivin protein encoded by the baculoviral IAP repeat containing 5 (BIRC5) gene is mainly expressed during the G2-M phase of the cell cycle where it regulates cytokinesis and chromosomal segregation [23]. Moreover, Survivin can counteract apoptosis by downregulating the activity of effector caspase-3 and -7, e.g., promoting resistance against radiation and other stressors [24]. Due to its pro-survival/anti-apoptotic activity, Survivin is highly expressed in most cancer tissues and is often associated with poor treatment outcomes [25]. Generally, high levels of Survivin suppressed or prevented the apoptosis initiation in malignant cells [24,26]. Furthermore, other studies showed that Survivin can act as a cytoprotective agent in non-malignant tissues such as the auditory system of the human cochlea and kidney [27,28]. In this context, Survivin is discussed as an adjusting therapeutic target to prevent apoptotic processes in injured tissue regions, as well as a crucial factor in transplantation medicine [29,30].

Regarding the lack of detailed knowledge about aSiNP-induced cell death, the objective of this study was to analyze the molecular effects of monodisperse amorphous silica nanoparticles (aSiNP) on cell models mimicking the alveolar-capillary barrier of the lung. By molecular characterization of endothelial lung cell line H441 and epithelial cell line ISO-HAS-1, we provided a model system suitable for in vitro nanotoxicity studies. Our experimental workflow, including state-of-the-art NP characterization, in vitro/in vivo imaging (fluorescence and electron microscopy), assessment of cell viability, as well as molecular pathway analyses, allowed us to give the first evidence that the anti-apoptotic protein Survivin acts as cytoprotective resistor against nanotoxicity. An analysis of healthy human and mouse lung tissue supports the clinical relevance of our in vitro data and the introduced model system as a first step system for further research projects.

## 2. Materials and Methods

### 2.1. Nanoparticles, Antibodies (Ab), and Reagents

Aqueous dispersions of acidic colloidal silica particles were purchased (aSiNP15: NexSil20A; aSiNP125: NexSil125A, Nyacol Nano Technologies, Inc., Ashland, MA, USA; aSiNPL: LudoxTM-40 Sigma Aldrich, Munich, Germany). Ab were Polyclonal rabbit and monoclonal mouse α (anti)-survivin (NB-500-201/NB-500-205, Novus Biologicals, Littleton, CO, USA); anti-ß-Actin (A2066; Sigma Aldrich); polyclonal rabbit α-phospho-AKT (9275, Cell Signaling); monoclonal rabbit α-AKT (4691; Cell Signaling, Danvers, MA, USA); monoclonal α-E-cadherin (610182, BD Transduction Laboratories); polyclonal rabbit α-p44/42 MAPK [pErk] (Thr202/Tyr204) (9212; Cell Signaling); polyclonal rabbit α-ZO-1 (40-2200; Invitrogen, Waltham, MA, USA); polyclonal rabbit α-cleaved Caspase-3 (9661, Cell Signaling); monoclonal mouse α-TTF-1 (M3575; Dako, Glostrup, Denmark). Appropriate secondary antibodies conjugated to HRP-, Cy3, or FITC were purchased from Sigma Aldrich (Munich, Germany) and Santa Cruz Biotechnology (Heidelberg, Germany). MEK inhibitor UO126 (Cell Signaling).

### 2.2. Characterization of Amorphous Silica Nanoparticles (aSiNPs)

Amorphous silica nanoparticles NexSil20A (aSiNP15), NexSil125A (aSiNP125), LudoxTM-40 (aSiNPL) were acquired commercially (Sigma Aldrich, Munich, Germany). In addition to SiO_2_, the used dispersion of SiNPs contains Na_2_O and has a pH of 10. The pristine nanoparticles without surface coating were characterized with respect to shape, size, and size distribution in dry state as well as in water-based solutions.

### 2.3. Transmission Electron Microscopy (TEM)

Imaging was performed using a Philips EM420 as described before [31].

### 2.4. Physicochemical Characterization Methods

The zeta potential was determined by the use of Zetaziser Malvern Zetasizer Nano ZS) with an appropriate device and parameter set up to examine the nanoparticle characteristics in regard to the aggregation behavior in different solutions [31]. For these measurements, particle stocks (600 mg/mL) were diluted in filtered MilliQ water (Millipore Billerica, Burlington, MA, USA) to final concentrations of 4.4 × 10^13^ NP/mL (600 µg/mL). Particle size, pH, and zeta potential were measured for all particles (Appendix A).

### 2.5. Endotoxin Detection in aSiNPs Dispersions

All aSiNP dispersions were analyzed for Gram-negative bacterial endotoxin contaminations by the QCL-1000^®^ Endpoint Chromogenic LAL Assay (Lonza Verviers, S.p.r.l., Verviers, Belgium) according to the manufacturer’s recommendations.

### 2.6. Preparation of aSiNPs for Apical Exposure

To avoid nanoparticle aggregation of commercially purchased aSiNPs (see Section 2.1), predilutions of the aSiNP dispersions were prepared in pure water (Aqua ad iniectabilia, Braun Melsungen AG, Melsungen). Due to nanoparticle aggregation in serum-containing medium, serum-free medium was used during the 4 h exposure. All dilutions were applied 1:10 in serum-free RPMI medium to the cells (10 µL NP-dispersion + 90 µL RPMI). Final concentrations of 10^10^ NP/mL (0.6 µg/mL), 10^11^ NP/mL (6 µg/mL) 10^12^ NP/mL (60 µg/mL), 10^13^ NP/mL (600 µg/mL) cover a range of non-toxic, sub-toxic and toxic concentrations and were chosen according to preliminary toxicity test performed for this project. After an exposure time of 4 h, the cells were either evaluated or washed twice with RPMI medium and cultivated for an additional 20 h period. After 4 h and 24 h recovery cell cytotoxicity was studied by various assays. In the context of the performed experiments, the additional incubation in the serum-containing medium was adjusted to the experimental requirements, described in detail in the figure captions.

### 2.7. Nanoparticle Uptake Studies

Cells were seeded on pre-coated coverslips (NUNC—Thermo Fisher Scientific, Dreieich, Germany) and cultivated until confluency was reached. Cells were exposed to 10^12^ NP/mL (aSiNP) in serum-deprived medium for 2 h at 37 °C. Cells were fixed in 3.7% (*v*/*v*) paraformaldehyde in PBS for 20 min at RT. Preparation for transmission electron microscopy analysis was performed as follows: Cells were fixed in 1% (*w*/*v*) osmium tetroxide (2 h, RT) and dehydrated in ethanol. Cells were passaged through propylene oxide, and then samples were embedded in agar-100 resin (PLANO, Wetzlar, Germany). Polymerization was acquired at 60 °C for 48 h. Resin block was cut into ultrathin sections using ultra microtome (Leica Microsystems, Wetzlar, Germany). Slices were placed onto copper grids for staining with 1% (*w*/*v*) uranyl acetate in alcoholic solution and lead citrate. Further analysis was performed by use of TEM technology (see Section 2.3).

### 2.8. Cultivation of Model Cell Lines ISO-HAS-1 and H441

Cell lines used for the applied cell model were acquired commercially from ATCC, ATCC-HTB-174, Promochem, Wesel, Germany. Cells were seeded according to their proliferation rate from confluent culture flasks into 96-well plates with 1.6 × 10^4^ cells/well for human microvascular endothelial cell line ISO-HAS-1 and with 3.2 × 10^4^ cells/well for human adenocarcinoma cell line H441. Cells were cultivated for 24 h (37 °C, 5% CO_2_) prior to SiNP treatment in RPMI 1640 medium (Gibco, Billings, MT, USA) containing L-glutamine, 10 % FCS, and Pen/Strep (100 U/100 µg/mL). The presence or absence of mycoplasma was examined using the commercial detection kit from Venor GeM Advance (Minerva Biolabs, Berlin, Germany). Cell counting was performed via Casy Cell Counter and Analyzer TT (Innovatis, Magdeburg, Germany).

### 2.9. Plasmids

The eukaryotic expression constructs for GFP and Survivin–GFP-tagged versions have been reported elsewhere [32]. The pGL3-luciferase reporter construct, pGL3Surv, containing the human Survivin promoter (nucleotides: –628 to +21), has been described [33]. Generation of recombinant retroviral particles and transduction was carried out as described [34,35].

### 2.10. Cells Transfection and Luciferase Assay

Stable expression of Survivin–GFP in H441 was established using transfection technology and subsequent fluorescence-activated cell sorting over two rounds described elsewhere [34]. Luciferase assays were performed as specified [36]. Luciferase reporter assays were performed in three independent experiments, each including a triplicate approach.

### 2.11. Apoptosis and Viability Assays

Caspase-3 hydrolysis was quantified using a fluorogenic substrate and via detection of cleaved caspase-3 using immunoblot-based analysis [36]. LIVE/DEAD™ Viability/Cytotoxicity Kit (Life Technologies, Darmstadt, Germany) was performed by supplier instructions to visualize loss of membrane integrity. Imaging was performed with digital Axiocam CCD camera (Carl Zeiss, Jena, Germany). Cell viability was examined using an ATP-dependent assay (CellTiter-Glo™ Viability Assay kit, Promega, Walldorf, Germany), resulting in a colorimetric change proportional to the number of viable cells. In addition, colorimetric MTT (3-(4,5-Dimethylthiazol-2-yl)-2,5-diphenyltetrazolium bromide) assay was performed to measure the metabolic activity, which correlates with the cell viability. Metabolic active cells cause a reduction in the MTT molecule resulting in a color change from a yellow tetrazole to a purple formazan dye (Sigma-Aldrich). All data were normalized to untreated control samples for each time point and statistically analyzed as reported in each figure caption.

### 2.12. Origin of Human and Mouse Tissue

Healthy human lung tissue was obtained from tumor patients at the University Hospital of Frankfurt. Studies of human tissue biopsies were performed according to the requirements of the local ethics committee, and informed consent was obtained in accordance with the Declaration of Helsinki. Four-week-old female NMRI nu/nu mice (Charles River Laboratories, Sulzfeld, Germany) were used. Animals were kept at 22 ± 1 °C under a 12:12 h light–dark cycle, and fed chow and water ad lib. Experimental protocols were approved by the local Animal Care and Use Committee. 

### 2.13. Tissue Preparation and Immunohistochemistry (IHC)

Human and murine lung tissues were fixed with formalin and embedded in paraffin (FFPE) prior to preparation for IHC as described [37]. Survivin was visualized using a polyclonal α-Survivin Ab, diluted 1:100. TTF-1 was applied with a 1:50 Ab dilution of monoclonal α-TTF-1 Ab.

### 2.14. Microscopy and Fluorescence Imaging of Cells

Observation, image analysis, and quantification of protein localization were performed as described [32]. Twelve-bit black and white images were captured using a digital Axiocam CCD camera (Carl Zeiss, Jena, Germany). Nuclei were stained with 1.25 µg/mL Hoechst 33,342 for 20 min (Sigma Aldrich, Munich, Germany). At least 200 fluorescent cells from three separate images were examined in three independent experiments.

### 2.15. Protein Extraction, Immunoblot Analysis, and Immunofluorescence

Treated cells were harvested and washed once in ice-cold PBS (1500 rpm, 4 °C, 5 min). Cells were sonicated in ice-cold modified extraction buffer to prepare lysates for immunoblot analysis [34,37]. Prior to application onto an SDS gel, lysates were diluted to similar protein concentrations and heated at 95 °C, 300 rpm for 5 min, with 6× dissociation buffer. Equal loading of lysates was controlled by detecting the housekeeping gene Actin. Immunoblotting was performed as described [36]. Detailed protocol for immunofluorescence has been described elsewhere [32,35].

### 2.16. Statistical Analysis

For experiments stating *p*-values, an unpaired *t*-test was performed. Unless stated otherwise, *p*-values represent data obtained from three independent experiments performed in triplicates. *p*-values < 0.05 were considered as significant (* *p* < 0.05; ** *p* < 0.01; *** *p* < 0.001, **** *p* < 0.0001). Statistical analysis and graph setup were performed using Prism9 (Graph Pad Software).

## 3. Results

### 3.1. ‘Characterization of the Nano–Bio Interface I’—Nano-scaled Amorphous Silica Nanoparticles

As the physico-chemical properties of NPs are critical determinants of NP-induced biological effects, we performed state-of-the-art characterization of the aSiNPs used in our study, aSiNP15/125 (aSiNP15/aSiNP125) and LudoxTM-40 (aSiNP L), analyzing morphology and shape, surface area, zeta potential, size, and stability (Figure 1, Appendix A).

(Cryo-)transmission electron microscopy (TEM) and dynamic light scattering (DLS) measurements revealed the stable size of the used aSiNPs in a dry state as well as in solution (see Appendix A). The hydrodynamic diameter remained constant, 33 nm for aSiNP15 and aSiNP L, or 125 nm for aSiNP125. Furthermore, all aSiNP retained a constant negative zeta potential in a serum-free cell culture medium, which can stabilize the suspensions via repulsive forces [38].

Importantly, since NP aggregation often precludes determining whether biological effects are triggered by individual NPs or NP aggregates (“microparticles”), time-dependent DLS measurements were performed (Appendix A). Distribution curves showed no significant aggregation when analyzing aSiNP dispersions in water, or physiological solutions such as phosphate-buffered saline (PBS, 540 mM salt in total) or serum-free cell culture medium (RPMI-1640; 139 mM salt in total, Appendix A). Furthermore, aSiNP15 and aSiNP L retain their nanoparticular size in water, PBS, and serum-free cell culture medium over time (Appendix A). Before all experiments, the absence of endotoxin contamination in the aSiNP preparations was controlled by an enzyme-linked immunosorbent assay ELISA (see Section 2.5).

### 3.2. ‘Characterization of the Nano–Bio Interface II’—A Biological Model to Mimic the Alveolar-Capillary Barrier of the Lung 

NPs can penetrate the body by different entry routes, such as the lung by passing the alveolar-capillary barrier (Figure 2A). Industrial exhaust or particulate matter products can enrich the air with harmful (nano)particles that can inadvertently enter the epithelial cells of the lung and further into the endothelial compartments, posing potential health risks to humans [39]. Thus, the alveolar-capillary barrier is an important part of the nano–bio interface and the physical defense system. In this context, studies of potential nanotoxic effects on the alveolar-capillary barrier are mandatory. However, this is rather difficult to access in vivo. Several models have been developed to mimic human lung tissue, such as different animal models, monolayer 2D cell cultures or 3D spheroid cultures, and organoids, to circumvent the poor availability of primary human samples [40].

Here, we introduce a dual cell line model that should mimic the alveolar-capillary barrier in vitro [41]. It consists of a human epithelial (adenocarcinoma) lung cell line with characteristics of type II pneumocytes and Clara cells (NCI H441) and the human microvascular endothelial cell (MEC) line ISO-HAS-1, mimicking the properties of alveolar lung tissue and functioning as a representative of endothelial-blood-vessel-forming cells (Figure 2A) [31,42]. Type II pneumocytes were chosen above type I pneumocytes because they produce the surfactant protein layer protecting and maintaining the alveolar stability important for physical defense [43,44]. Although both cell lines originate from tumor tissue, their use as in vitro models for functional, healthy lung tissue has been already suggested in previous studies [45].

Our model system was characterized molecularly to prove its suitability as a model system of the lung. Therefore, immunofluorescence/histochemical staining of specific lung cell markers and general differentiation markers was utilized. First, we performed marker profiling in human lung tissue to demonstrate physiological relevance. Here, specific expression of Cluster of Differentiation 31 (CD31) in the cellular membrane of endothelial, alveolar cells (Figure 2B, left), as well as nuclear expression of the thyroid transcription factor-1 (TTF-1) in epithelial cells (Figure 2B, right), could be revealed. Next, molecular marker profiling confirmed the epithelial character of the lung cell line H441 by positive staining for E-cadherin and the epithelial cell adhesion molecule EpCAM (Figure 2C, upper panel). In contrast, the endothelial cell line ISO-HAS-1 was characterized as E-cadherin/EpCAM negative (Figure 2C, lower panel). Furthermore, ISO-HAS-1 could be distinguished from H441 by positive staining for the endothelial-specific adhesion molecule CD31. In order to further prove the suitability of our in vitro system, we showed that the lung (and thyroid)-specific transcription factor TTF-1 (thyroid transcription factor-1) is specifically expressed in epithelial H441 cells, but not in endothelial ISO-HAS-1 cells (Figure 2C). Since TTF-1 is essential for the expression of lung surfactant proteins, such as Surfactant protein-A (SP-A), and is thus crucial for maintaining a healthy morphology as well as host defense mechanisms of the lung [46,47], we also analyzed our model cell lines for SP-A expression. Consistent with our findings for TTF-1, SP-A is expressed exclusively in H441 cells, whereas ISO-HAS-1 cells were characterized as TTF-1/SP-A-negative (Figure 2C). Additionally, staining of ZO-1 (Zonula occludens-1, Tight junction protein-1) was performed to visualize the presence of tight junctions, which are critical for functional and protective cell barriers [48].

In summary, we here revealed several similarities between healthy human lung tissue and our characterized two-cell model, suggesting that this (tumor-derived) system is suitable for in vitro nanotoxicity studies mimicking the alveolar-capillary barrier of the (healthy) lung.

### 3.3. ‘Effects at the Nano–Bio Interface’—aSiNPs Induce Cell Death upon Exposure to Alveolar-Capillary Barrier Model Cells

After confirming that the used aSiNP preparations remain stable under physico-chemical cell culture conditions (see Section 2.2 and Section 2.4), we next investigated the effects of aSiNPs on our lung model cells H441 and ISO-HAS-1. First, to visualize cellular surface interactions and the potential uptake of aSiNP, we treated H441 cells with increasing amounts of fluorescent aSiNP (Kisker red, 50 nm, Nylon) in a serum-deprived medium (Figure 3A). Because serum proteins can cover nanoparticles with a protein corona, serum-free conditions were chosen to retain aSiNP’s full hazardous potential for all experiments (see Section 2.6) [49]. Interestingly, Kisker NPs could be detected at the outer cell membrane, as well as internalized in the cell, and the amount of interacting/internalized aSiNPs increased with the used NP dose.

To obtain a more detailed picture of the aSiNP–cellular interaction, which might also allow conclusions about biological effects, confluent H441 monolayers were treated with 10^12^ aSiNP15 NP/mL (in medium, 2 h, see Section 2.7), and fixed and analyzed by transmission electron microscopy (Figure 3B). Interestingly, in the presence of serum, aSiNP15s were primarily located extracellularly. Since serum proteins can form a protein corona around nanoparticles, especially around pristine nanoparticles with a negative surface charge like the used aSiNP15 [49,50], we examined the uptake of aSiNP15 in the absence of fetal calf serum (FCS) (Figure 3C,D). Here, aSiNP15s were detectable on the cell surface, but also freely dispersed in the cytoplasm, and localized in membrane-bound endocytic vesicles, indicating the internalization of aSiNP15s (Figure 3C,D, arrows). Detailed examination of representative TEM images also revealed a more irregular shape and/or a darker border covering the aSiNPs incubated with serum (Figure 3B, lower panel). This NP-surrounding layer vanished in the absence of serum (Figure 3C,D, lower panel), and thus can be interpreted as protein corona covering the NPs. Similar results were obtained in ISO-HAS-1 cells and for other aSiNP types, such as aSiNP L (Appendix A). Based on these findings, subsequent aSiNP exposure experiments were conducted in a serum-free medium to prevent the binding of serum proteins to the nanoparticles, which might significantly affect their physico-chemical/biological properties.

As a next step, we performed cell viability measurements using aSiNP15 in our dual cell line model revealing dose- and time-dependent toxicity (Figure 4). Apical exposure of epithelial (H441) or endothelial (ISO-HAS-1) cell lines to 10^13^ NP/mL resulted in cell death 4 h after NP exposure, as revealed by a viability assay based on the cell membrane integrity (Figure 4A). Lower particle concentrations (10^12^ NP/mL) led to a partial loss of the membrane integrity after 24 h (Figure 4A, lower panel). However, both cell types responded similarly to aSiNP15 treatment.

These results were confirmed by measuring the metabolic activity (ATP content) of viable cells (Figure 4B). ISO-HAS-1 as well as H441 cells revealed decreased vitality after treatment with NP concentrations above 10^10^ NP/mL. Particle concentrations of 10^12^ NP/mL showed a constant decrease in viability over the treatment period of 24 h in total, ranging from 69.6% (4 h) over 44.2% (8 h) to 5.8% (24 h) in ISO-HAS-1 cells relative to untreated controls (Figure 4B, upper panel). Comparable results were obtained for epithelial cells H441 (79.6%, 4 h; 47.2%, 8 h; 21.5%, 24 h). Higher particle concentrations (10^13^ NP/mL) increased toxicity at early time points in both cell types: viability of ISO-HAS-1 cells decreased to 34.5 %, and of H441 cells to 17 % after exposure to 10^13^ NP/mL for 4 h, and declined further with treatment time (Figure 4B).

Additionally, morphological changes were evaluated over the incubation period of 24 h (Appendix A). Low aSiNP15 concentrations (10^10^ NP/mL) had no impact on either ISO-HAS-1 or H441 cells over 24 h. However, prolonged incubation with high NP concentrations (10^13^ NP/mL) led to cell damage, indicating apoptotic and/or necrotic phenotypes (Appendix A). Interestingly, whereas aSiNP L (with comparable size to aSiNP15) treatment resulted in comparable toxic effects (Appendix A), aSiNPs of a larger size (aSiNP125) resulted in significantly less apoptotic and/or necrotic cells under the same treatment conditions (Appendix A). These findings were further consolidated with uptake studies, which revealed that aSiNPs of greater size (aSiNP125) were also taken up by the cells without the induction of an apoptotic phenotype (Appendix A).

To independently prove that aSiNP15 treatment results in the induction of apoptosis, we quantified the expression of cleaved caspase-3, a major effector caspase, which is activated by cleavage during apoptosis initiation. First, immunohistochemical staining of cleaved caspase-3 in our dual cell line lung model revealed that exposure to 10^12^ NP/mL or higher concentrations triggered apoptosis after 4 h of treatment in both cell types (Figure 4C). Dose- and time-dependent induction of apoptosis was confirmed by an in vitro caspase-3 activity assay (Figure 4D). Whereas treatment with 10^12^ NP/mL likewise resulted in significant activation of caspase-3 and, thus, apoptosis induction after 4 and 24 h, lower concentrations of aSiNP15 (here 10^10^ NP/mL) just marginally activated the caspase-3 cascade after an elongated treatment period (24 h). Consistent with our previous data exposure, 10^12^ aSiNP125/mL had no comparable apoptotic effect under the same treatment conditions [49]. These data indicate that aSiNP-induced toxicity could critically depend on the particle size, besides the particle concentration.

### 3.4. ‘Signaling at the Nano–Bio Interface’—High Concentrations of aSiNP15 Attenuate Pro-Survival Pathways

Next, our study aimed at providing insights into the underlying molecular mechanisms of aSiNP-induced cell death. Regulation of cellular survival and death involves a complex signaling network in which several phosphorylating kinases play key roles. For example, phosphatidylinositol 3-kinase (PI3K)/Akt and ERK (extracellular-signal-regulated kinases; also known as mitogen-activated protein kinases, MAPK) signaling are known pro-survival pathways in lung and endothelial cells, and their contributions to apoptosis regulation are well documented [51].

To examine if aSiNP15-induced toxicity depends on PI3K/Akt and/or ERK signaling, immunoblot analyses were performed profiling the (activation-dependent) phosphorylation of both kinases (Figure 5). Interestingly, PI3K/Akt and ERK signaling was attenuated in a dose- and time-dependent manner: whereas exposure to 10^12^ NP/mL reduced the levels of phosphorylated Akt and ERK over 6 h of treatment (Figure 5A and Appendix A), treatment with 10^10^ NP/mL did not affect phosphorylation nor the activation of either Akt or ERK signaling (Figure 5B).

To further confirm the relevance of the ERK signaling pathway for aSiNP-mediated toxicity, the MEK inhibitor UO126 was utilized to suppress ERK1/2 signaling (Figure 5C). Importantly, inhibitor treatment further increased aSiNP15-induced toxicity, likewise, in a time- and dose-dependent manner. The combination of MEK inhibition with high aSiNP15 concentrations (10^12^ NP/mL) resulted in a significant decrease in cell viability after 4 h. Interestingly, pre-treatment with the MEK inhibitor sensitized cells to aSiNP15-induced toxicity as revealed by a significant reduction in cell viability even when treated with sub-lethal aSiNP15 concentrations of 10^10^ or 10^11^ NP/mL for extended incubation times of 8 h or 24 h.

In sum, these results strongly support the hypothesis that aSiNP-induced toxicity is mediated by the attenuation of PI3K/Akt and ERK pro-survival signaling.

### 3.5. ‘Survival and Death at the Nano–Bio Interface’—aSiNP-Induced Cell Death Is Regulated by the Anti-Apoptotic Protein Survivin

Within the complex network of regulating cell death and survival, a plethora of individual molecular interactions govern the mutual influence of single proteins in diverse signaling cascades. One molecular actor involved in these pathways is a member of the inhibitor of apoptosis protein (IAP) family, Survivin/BIRC5. Interestingly, previous studies have suggested that PI3K/Akt and ERK signaling might stimulate Survivin expression, while suppression of these pathways attenuates Survivin protein levels, subsequently resulting in apoptosis induction [37,52,53,54].

Thus, we analyzed the expression of the anti-apoptotic protein Survivin after aSiNP15 treatment on the protein and the transcriptional level (Figure 6). Remarkably, immunoblot analysis revealed that Survivin expression decreased in ISO-HAS-1 and H441 cells treated with 10^12^ NP/mL over an incubation period of up to 6 h (Figure 6A,B and Appendix A). Again, lower aSiNP15 concentrations (10^10^ NP/mL) did not significantly affect Survivin expression (Figure 6D,E). Interestingly, long-term incubation over 24 h revealed cell line-dependent effects concerning protein expression. Whereas high aSiNP15 concentrations (10^12^ NP/mL) led to decreased Survivin levels after 24 h in both cell lines (Figure 6C), the sublethal aSiNP dose (10^10^ NP/mL) reduced the amount of Survivin protein in H441 cells, but not in ISO-HAS-1 (Figure 6E). On the contrary, Survivin expression appeared to be increased in ISO-HAS-1 cells after 24 h of incubation with low concentrations of aSiNP15 (10^10^ NP/mL, Figure 6F).

To investigate whether Survivin downregulation also occurs on the transcriptional level, H441 cells were transfected with a luciferase reporter plasmid containing the Survivin promoter sequence (Figure 6G, Appendix A). The reporter assay indeed confirmed that high particle concentrations (10^11^ and 10^12^ NP/mL) induce a significant decrease in the Survivin promoter activity after 4 h of treatment. A prolonged incubation period (24 h) further decreased Survivin transcription, while low particle concentrations (10^10^ NP/mL) had no repressive effect, which is in line with the immunoblot results (Figure 6C,F).

Importantly, to further confirm Survivin`s protective role against aSiNP-induced toxicity, we analyzed whether overexpression of Survivin might be able to counteract cell death after aSiNP15 treatment (Figure 7). First, H441 cells were retrovirally transduced with the Survivin–GFP (green fluorescent protein) fusion protein or internal ribosome entry site (IRES)–GFP constructs, and selected for GFP expression. Immunofluorescence microscopy of stably expressing cells confirmed characteristic sub-cellular localization patterns of Survivin during mitosis and in interphase cells (Figure 7A). Ectopic expression and functionality of the Survivin-GFP construct was verified by fluorescence microscopy (Figure 7A). Next, cell lines were challenged with low (10^10^ NP/mL) and high doses (10^12^ NP/mL) of aSiNP15 for up to 24 h (Figure 7B–D). In comparison to the control cells (H441), Survivin-overexpressing cells (H441 Surv) showed a significant decrease in apoptosis induction (Figure 7D). Furthermore, these cells kept their healthy phenotype upon aSiNP15 treatment, while shrinking and cell detachment from the culture dish occurred in wild-type cells for the same treatment conditions (Figure 7B and Appendix A). However, aSiNP15-induced cell death after long-term incubation could not be prevented by Survivin overexpression but was significantly decreased in comparison to wild-type cells (Figure 7C).

Taken together, our data strongly suggest that aSiNP15-induced toxicity depends, at least in part, on the regulation of the anti-apoptotic protein Survivin. Further, the results indicate that a high expression level of Survivin potentially attenuates aSiNP15-induced toxicity.

### 3.6. ‘Probing Physiological Relevance of the Nano–Bio Interface’—Survivin Is Expressed in Healthy Mouse and Human Lung Tissue

Since our in vitro alveolar-capillary model suggests Survivin as a cytoprotective resistor against aSiNP-induced toxicity, it is mandatory to analyze its physiological relevance in vivo. Therefore, we used an established immunohistochemistry (IHC) protocol to analyze Survivin expression in human and murine lung alveolar cells (ATII) (Figure 8A,C) [34,37]. Again, the thyroid transcription factor-1 (TTF-1) was used as a common marker for ATII-lung cells. The IHC staining visualized specific staining of Survivin consistent with TTF-1-labeled epithelial lung cells (Figure 8B,D). Tissues from different origins also differ in their staining pattern, which has to be considered for the model system, which should mimic the target tissue: lung tissue from mouse origin revealed no TTF-1-positive cell layer of epithelial cells and showed no consistency with Survivin-positive cells (Figure 8C,D), in contrast to the human lung tissue (Figure 8A,B). Of note, no IHC signal was detectable upon omission of the primary α-Survivin Ab or pre-absorption of the α-Survivin Ab with recombinant human Survivin–GFP protein, confirming staining specificity.Conclusively, our data confirm that Survivin is expressed in healthy lung tissue, and thus Survivin’s cytoprotective role against aSiNP-induced toxicity is also relevant for animal-based research and its clinical transfer to human disease, but tissue differences should be considered in the experimental setup.

## 4. Discussion

Nanoparticles, as easy and cheap to synthesize materials, are very popular due to manifold modification opportunities. In particular, silica nanoparticles are easy to modify and are frequently used in industry, most notably in cosmetic applications, food processing, and the medical industry as part of drug formulations [3,4,10,12,55]. The growing number of nanoparticle-containing products as well as nanoparticle waste makes it necessary to monitor their possible harmful effects and simultaneously examine treatment options. In this context, the demand for suitable lung models regarding the complexity of tissue samples, ethical issues, and the experimental handling of animal models is increasing. We therefore compared specific differentiation markers of human lung tissue to those of mouse lung and cell culture models. For the first time, we showed that tissue cells of the human lung express the inhibitor of apoptosis protein, Survivin, which is critical for mitosis and proliferation [31,56,57,58,59]. Furthermore, we implemented cell culture models that reflect the necessary characteristics of lung tissue cells. Monoculture cell line models, consisting of the human epithelial lung carcinoma cell line H441 and human microvascular endothelial cell (MEC) line ISO-HAS-1, reproduce necessary properties of differentiated lung tissue to a good extent and can, therefore, serve as good model systems for lung tissue. On a molecular level, adhesion molecules are necessary for tissue integrity and, consequently, for their functionality, like EpCAM or E-cadherin for epithelial lung tissue or CD31 for endothelium [60,61]. Consequently, the differentiation status of the epithelial cell line H441 and endothelial cell line ISO-HAS-1 was distinguished by tissue-specific differentiation markers. H441 expressed epithelial-specific markers α-E-cadherin and α-EpCAM pivotal for the generation of epithelial cell–cell contact and barrier formation, as well as lung-specific markers α-SP-A and α-TTF-1 [62,63,64,65,66,67]. The endothelial differentiation marker CD31 was expressed exclusively by the ISO-HAS-1 cell line, which proves that the two selected cell lines are highly differentiated and suitable for establishing simplified lung culture models [68].

The pro-survival stimulating properties of IAP family member Survivin could be a conceivable approach in the treatment of tissue damages, e.g., regarding the possible beneficial use of Survivin in transplants [29]. Thus, Survivin is known to increase cancer cell proliferation and resistance against chemotherapeutics and radiation [23,24,69]. It was also demonstrated that Survivin plays a key role in protecting and restoring the auditory system after ototoxic effects have led to hearing loss [28]. Several studies showed that Survivin is expressed in a broad range of non-malignant tissues and stem cells, indicating the crucial role of Survivin as a pro-survival factor in increasing resistance and proliferation [70,71]. These cytoprotective mechanisms may likewise be used against nanoparticular stress as proposed in this work. 

In comparison to the corresponding mouse model, the dual lung model, based on cell lines, is well suited to reflect the specific properties of lung tissue cells and to simultaneously enable simplified and cell-specific analyses. However, type I pneumocytes, representing the majority of the lung surface tissue cells, should be considered for future research approaches on lung model systems [44]. On the basis of the used cell model, sublethal concentrations of aSiNP15 did indeed not affect the two cell lines, whereas higher particle doses resulted in morphological changes. We subsequently revealed decreasing cell viabilities over the treatment time, suggesting a time- and concentration-dependent toxicity effect. In addition, we found that a larger nanoparticle size reduced the toxicity of aSiNPs at constant material properties and concentrations, which, besides differences in their surface area, might be also attributed to a size-dependent protein corona formation [50]. However, Survivin expression was reduced by higher aSiNP15 concentrations, as well as the Survivin-enhancing prosurvival factor ERK. These results indicate that the MEK/ERK/Survivin axis is part of the defense mechanism against aSiNP toxicity. Survivin expression, location, and modification are modulated by several signaling pathways. To demonstrate that aSiNP toxicity is mediated by decreasing phosphorylation of ERK1/2, we made use of an MEK inhibitor to prevent the activation of ERK1/2 by MEK-mediated phosphorylation. In contrary to previous experiments the use of non-toxic concentrations of aSiNP15 led to significantly lower viabilities in combination with the MEK inhibitor. Therefore, the cytoprotective effect of survivin might be decreased due to reduced ERK1/2 activation by the inhibition of MEK-mediated ERK phosphorylation. The data suggest that suppressed phosphorylation of ERK may contribute to decreased Survivin expression by interfering with MAPK/ERK pro-survival signaling, in turn, leading to caspase-3 activation and apoptosis.

The effect of aSiNP on cells has been discussed in numerous publications, but the mechanisms are still not clearly understood [72]. The most commonly observed effects of aSiNP exposure include the induction of apoptosis through DNA damage or mitochondria-, TNF-α-, and NO-related signaling pathways [72,73]. Mechanistically, silanol groups (Si-OH) on the nanoparticle surface can interact electrostatically with the cell membrane [74]. Furthermore, their nucleophilic character can attack electrophilic carbonyl groups of proteins [75]. Since we used non-coated SiNP without the addition of any surfactant, potential unspecific effects caused by foreign ions and/or surface functionalizations should be negligible. In addition, signaling processes are likely to be triggered by the generation of reactive oxygen species (ROS) mediated by the cleavage of strained siloxane rings [74,75,76,77]. We showed that aSiNPs accumulate in the cells with increasing particle concentrations correlating with morphological changes and augmented cell toxicity. Whether Survivin can induce ROS-clearing processes under sub-toxic conditions, or if several different defense responses are addressed by Survivin, remains to be clarified.

Zhao et al. (2010) and others showed that Survivin is sufficiently induced by the PI3K/Akt/p70S6K1 pathway, thereby protecting cells from apoptotic stimuli [52,54]. Other studies described Akt/PI3K activation mediated by low NO levels as a Survivin-inducing pathway [28]. In comparison to untreated controls, the amount of phosphorylated Akt and ERK was not enhanced by non-toxic aSiNP15 concentrations in both cell types, underlining that these pathways are necessary for sustaining normal cell homeostasis [78]. Further, the PI3K/Akt pathway was less affected by aSiNP15 treatment in endothelial ISO-HAS-1 cells compared to the MEK/ERK pathway. However, H441 cells exhibited slightly less phosphorylated Akt, while ERK1/2 was highly phosphorylated after incubation with high particle concentrations, correlating with a decreasing Survivin expression.

Survivin is a common tumor marker aberrantly expressed in various cancer types [79,80,81,82]. However, overexpression of Survivin via transduction of H441 cells with a Survivin–GFP construct enhanced their resistance to aSiNP15 treatment. These data suggest that Survivin indeed serves as an active resistor against aSiNP15-mediated toxicity for non-necrotic particle concentrations. aSiNP-mediated stress signaling addresses the Ras/Raf/MEK/ERK pathways rather than the PI3K/Akt pathway in the endothelial cell line ISO-HAS-1. A crosstalk between these pathways is also conceivable, making it more complex to understand and utilize for the further development of clinical treatments [83].

## 5. Conclusions

Due to their wide range of applications, aSiNPs have become a constant companion in our everyday lives. Therefore, it is of tremendous importance to examine possible hazardous effects on the human system. The most common entry routes for aSiNPs are the epithelium of the skin, the gastrointestinal tract, and the alveoli of the lung. We can confirm that silica nanoparticles of the type aSiNP15 exert a concentration- and time-dependent negative impact on the viability of epithelial and endothelial cells. Based on the presented data, we suggest that Survivin protects cells against aSiNP-mediated toxicity. The mechanisms behind the empirical data include the MAPK pathway by MEK activation and subsequent ERK1/2 phosphorylation leading to enhanced Survivin levels and apoptosis inhibition. To underline the relevance of our study for (patho)physiological conditions, we demonstrated that Survivin is expressed in ATII-type lung cells, as well as in healthy human and murine lung tissue. Furthermore, we propose a well-differentiated dual cell model consisting of H441 and ISO-HAS-1 cells mimicking lung tissue in a simplified and transferable way usable for high throughput experiments prior to (pre-)clinical in vivo studies on human or mouse tissue of the lung.

## Figures and Tables

**Figure 1 nanomaterials-13-02546-f001:**
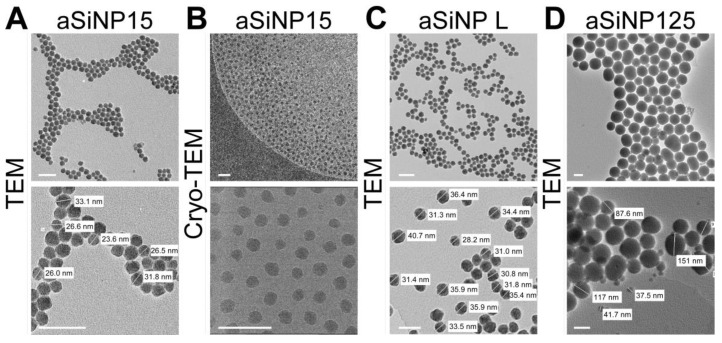
Characterization of the used amorphous silica NP system (aSiNP) by (Cryo-)transmission electron microscopy (TEM). Representative pictures of analyzed aSiNPs are shown in overview (upper row) and higher magnification (lower row): (**A**) aSiNP15 (TEM), (**B**) aSiNP15 (Cryo-TEM), (**C**) aSiNP L (TEM), and (**D**) aSiNP125 (TEM). Scale bars, 100 nm.

**Figure 2 nanomaterials-13-02546-f002:**
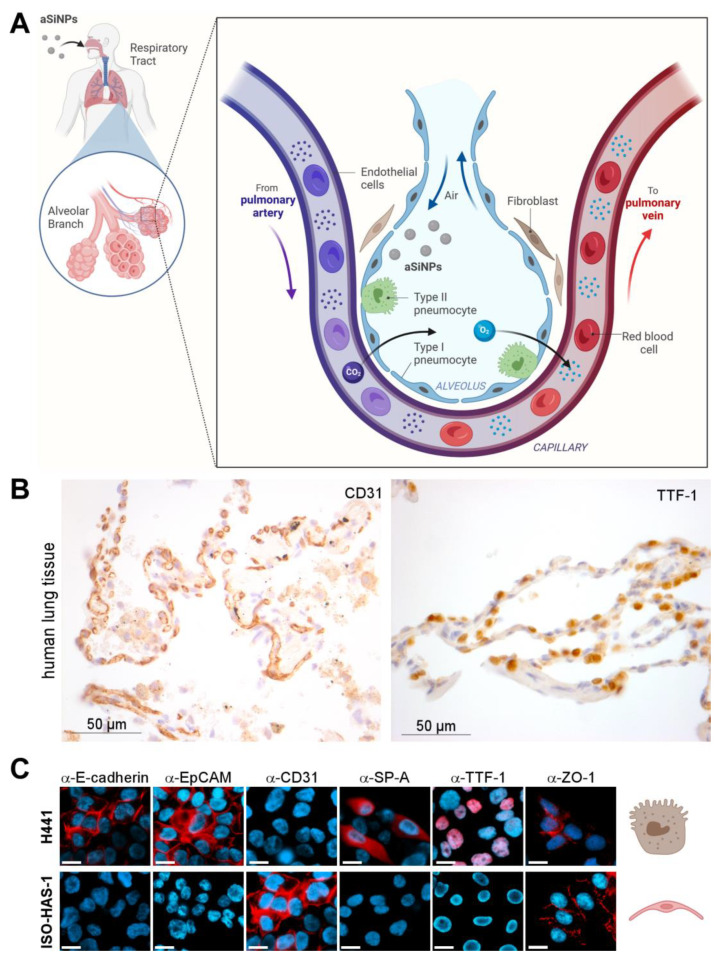
Biological model to mimic the alveolar-capillary barrier of the lung: (**A**) Illustration of a bronchial strain as the first protective cellular barrier against inhaled nanoparticle-containing air. (**B**) Light microscopic analysis of biomarker expression. Expression of CD31 and TTF-1 was analyzed in epithelial human lung tissue by immunohistochemistry with specific antibodies (brown). HE (hematoxylin/eosin) staining was applied to visualize tissue morphology (blue). CD31 surface marker was present on the membrane of epithelial cells in the human lung tissue, whereas TTF-1 was prominently expressed in cell nuclei. (**C**) Marker profiling of the H441 and ISO-HAS-1 dual cell line model was carried out by fluorescence microscopy after staining with specific antibodies (α-E-cadherin, α-EpCAM, α-CD31, α-SP-A, α-TTF-1, α-ZO-1) and the nuclear stain Hoechst-33342 (blue) to evaluate the cellular differentiation state. Scale bars, 10 µm. Respective cell types are symbolized by referring to (**A**). Created with BioRender.com.

**Figure 3 nanomaterials-13-02546-f003:**
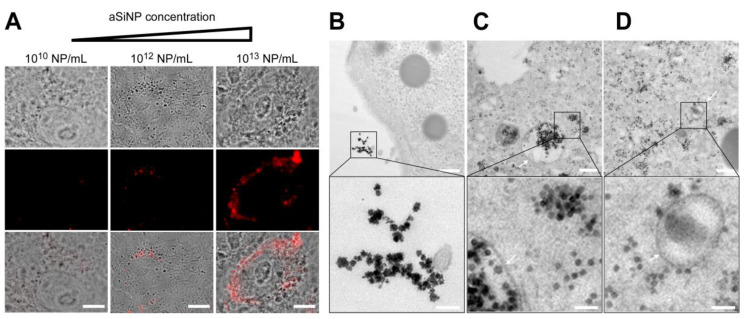
Interaction and internalization of aSiNPs by epithelial lung cells: (**A**) Immunofluorescence and light microscopy of H441 cells after incubation with increasing concentrations of fluorescent aSiNP (Kisker red) for 2 h in serum-deprived medium. Scale bars, 10 µm. (**B**–**D**) aSiNP15 nanoparticle uptake by H441 cells was visualized by TEM. H441 cells were treated with 10^12^ NP/mL for 2 h in serum-containing (**B**) or serum-free medium (**C**,**D**). Representative images are shown in overview (upper row), and in detail view (lower row) indicating the presence (**B**) or absence (**C**,**D**) of a protein corona around aSiNP15. Endocytic vesicles are marked by arrows. Scale bars, 500 nm (upper panel), and 100 nm (lower panel).

**Figure 4 nanomaterials-13-02546-f004:**
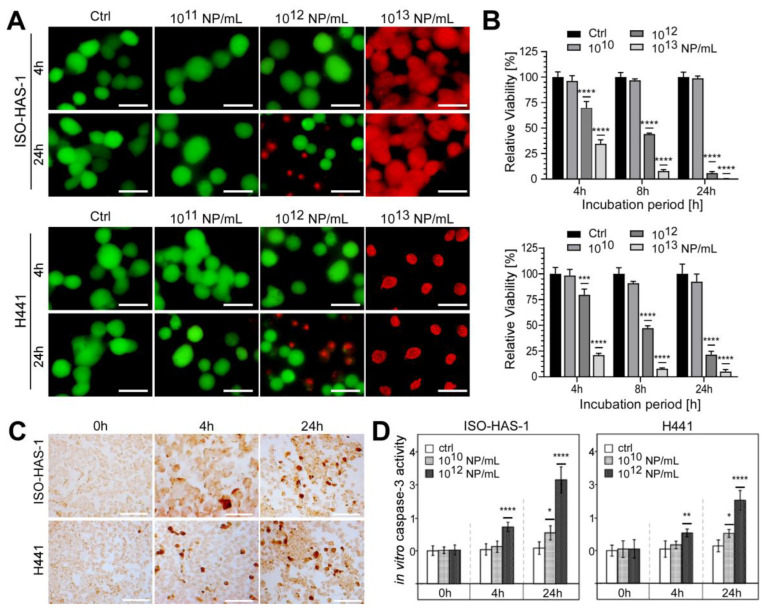
aSiNP15 treatment induced dose- and time-dependent toxicity on a dual alveolar-capillary barrier model. ISO-HAS-1 and H441 cells were treated with different particle concentrations in serum-free medium for 4 h; for later time points, treated cells were cultured for additional 20 h in serum-containing medium: (**A**) Live and dead staining analyzed by fluorescence microscopy. Green fluorescence marks viable cells; red fluorescence visualizes dead cells. (**B**) CellTiter-Glo™ Viability Assay. Data were normalized to untreated control samples for each time point. Unpaired *t*-test was used to compare treated cells to untreated control groups. Column, mean; bars, ±S.D. ***, *p* < 0.001, ****, *p* < 0.0001. (**C**,**D**) aSiNP15 induced apoptosis in ISO-HAS-1 and H441 cells via activation of caspase-3. (**C**) Light microscopic analysis of apoptosis marker caspase-3. Immunohistochemical staining of ISO-HAS-1 (upper panel) and H441 (lower panel) revealed expression of cleaved caspase-3 as a marker for apoptosis after 4 h of aSiNP15 treatment. Briefly, cells were treated with 10^12^ NP/mL for 4 or 24 h, fixed, stained for cleaved α-caspase-3, and visualized by light microscopy. Scale bars, 150 µm. (**D**) Quantification of in vitro caspase-3 activity confirmed time- and dose-dependent activation of apoptosis. Cleaved α-caspase-3 assay was performed with untreated and aSiNP15-treated cells. Cells were incubated for 4 h and 24 h with 10^10^ or 10^12^ NP/mL as indicated. Data were normalized to untreated control samples for each time point. Unpaired *t*-test was used to compare treated cells to untreated control groups. Column, mean; bars, ±S.D. *, *p* < 0.05, **, *p* < 0.01, ****, *p* < 0.0001.

**Figure 5 nanomaterials-13-02546-f005:**
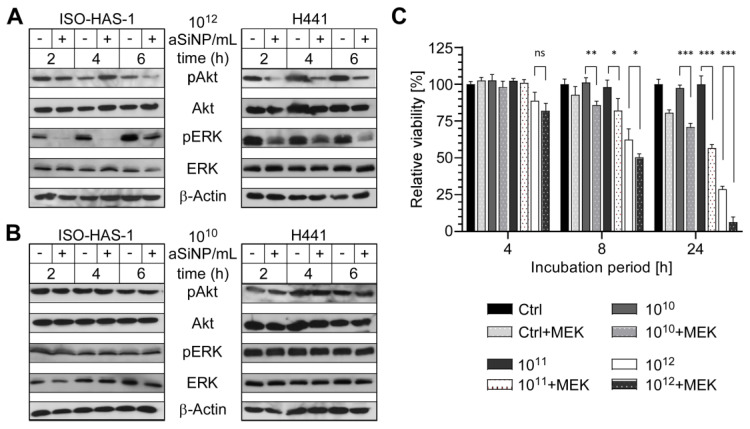
High concentrations of aSiNP15 attenuate PI3K/Akt and ERK signaling. (**A**,**B**) Activation of PI3K/Akt and ERK signaling pathways was analyzed by assessing the level of phosphorylated PI3K/Akt (pAkt) and ERK (pERK; Thr202/Tyr204) by immunoblot analysis. Whereas sub-toxic NP concentrations (10^10^ NP/mL) had no effect (**B**), pAkt and pERK decreased after incubation with 10^12^ NP/mL. ISO-HAS-1 and H441 were treated with 10^12^ NP/mL (**A**) or 10^10^ NP/mL (**B**) aSiNP15 (indicated by “+”), or in serum-deprived medium as control (indicated by “−“) for 2 h, 4 h, and 6 h. Proteins were detected by (phosphorylation-) specific antibodies, the respective non-phosphorylated proteins, and Actin served as loading controls. (**C**) Relevance of ERK signaling was confirmed by combined MEK (mitogen-activated protein kinase kinase) inhibitor treatment. After H441 cells were pre-treated for 2 h with or without the MEK inhibitor UO126 (10 µM), treatment with 10^10^, 10^11^, or 10^12^ NP/mL of type aSiNP15 was conducted for 4 h, 8 h and 24 h. Cell viability was measured via an MTT assay. Data were normalized for each time point to untreated controls. (*) Unpaired *t*-test was performed to compare MEK inhibitor-treated samples to samples with similar treatment without inhibitor. Columns, mean; bars, ±S.D. *, *p* < 0.05, **, *p* < 0.01, ***, *p* < 0.001, ns, not significant.

**Figure 6 nanomaterials-13-02546-f006:**
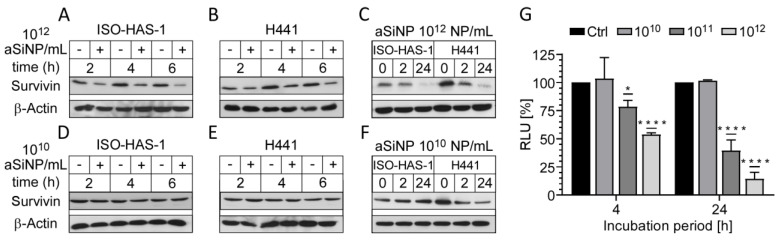
aSiNP-mediated toxicity was accompanied by dose-dependent downregulation of the anti-apoptotic protein Survivin: (**A**–**F**) Immunoblot analysis revealed dose-dependent downregulation of Survivin in response to treatment with aSiNP15. ISO-HAS-1 and H441 cells were treated with (“+”) 10^12^ NP/mL (**A**–**C**) or 10^10^ NP/mL (**D**–**F**) aSiNP15 or without particles (“−“) for 0 and 2 h in serum-free medium. For incubation times longer than 4 h, serum-containing medium was added after 4 h to prevent starvation-induced cell death. Proteins were detected by specific antibodies. β-Actin served as loading control. (**G**) Suppression of the Survivin promoter was revealed by a luciferase reporter assay following aSiNP15 treatment. Briefly, H441 cells were treated with 10^10^, 10^11^, and 10^12^ NP/mL for 4 h in serum-free medium. Likewise, for the 24 h time point, serum-containing medium was added after 4 h. After incubation, cells were lysed, and lysates were used for quantification of luciferase activity as described [36]. Data were normalized to untreated controls, and unpaired *t*-test was performed. Columns, mean; bars, ± S.D. *, *p* < 0.05, ****, *p* < 0.0001.

**Figure 7 nanomaterials-13-02546-f007:**
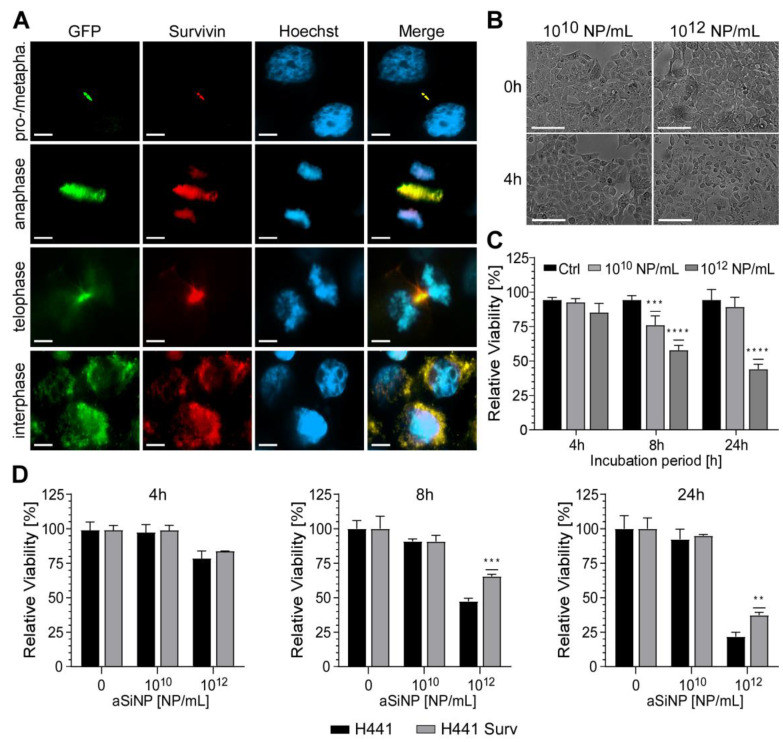
Overexpression of Survivin counteracts aSiNP15-induced cell death: (**A**) In stably Survivin-GFP-expressing H441 cells (H441 Surv), Survivin exhibits its characteristic localization during mitotic phases shown via immunofluorescence microscopy (as indicated). In prophase/metaphase, it localizes at chromatin sides. In anaphase, Survivin dissociates from chromatin and associates with the midzone spindle apparatus. In telophase, it localizes at the midbody, and disperses in the cytoplasm and/or in the nucleus during interphase. Scale bars, 10µm. (**B**) Morphological stability of H441 cells after treatment with different aSiNP15 concentrations at the beginning of treatment (0 h) and after 4 h aSiNP15 exposure. Cell morphology was visualized by light microscopy. Scale bars, 150 µm. (**C**) High doses of aSiNP15 induce cell death after long-term incubation. An MTT viability assay of stably expressing Survivin–GFP H441 cells was performed after 4 h, 8 h, and 24 h of aSiNP15 treatment as indicated. Data were normalized to untreated controls, and unpaired *t*-test was performed. Columns, mean; bars, ±S.D. ***, *p* < 0.001, ****, *p* < 0.0001. (**D**) CellTiter-Glo^TM^ viability assay of H441 Surv (grey) and control cells (H441, black) was performed after 4 h, 8 h, and 24 h of aSiNP15 treatment as indicated. Data were normalized to untreated controls, and unpaired *t*-test was performed. Columns, mean; bars, ±S.D. **, *p* < 0.01, ***, *p* < 0.001.

**Figure 8 nanomaterials-13-02546-f008:**
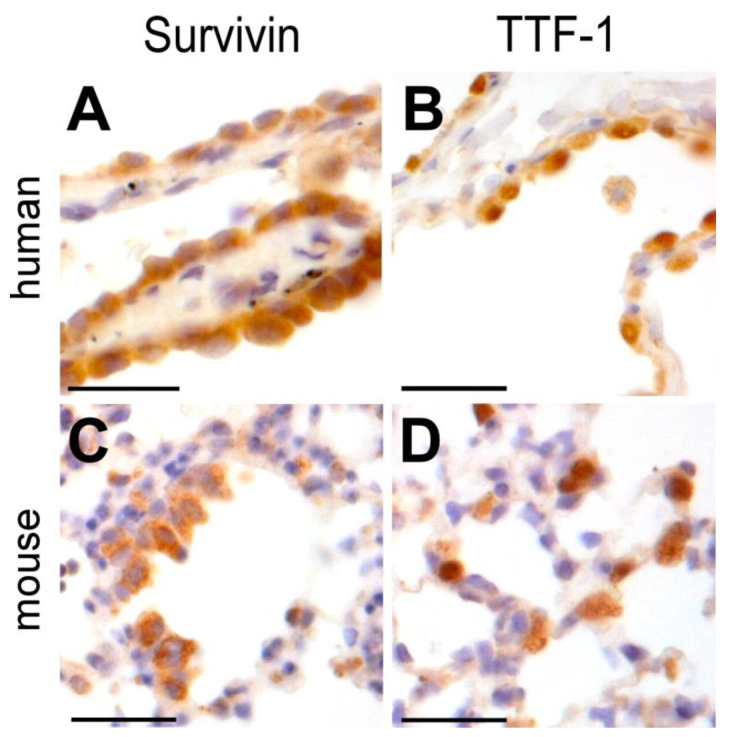
Survivin is physiologically expressed in epithelial cells of the human and mouse lung. Immunohistochemical staining of Survivin (**A***,***C**) and TTF-1 (**B**,**D**) in epithelial human and mouse lung tissue with specific antibodies (brown) and visualized by light microscopy. HE staining was applied to visualize lung tissue structure (blue). Survivin localized in the nuclei and cytoplasm of mouse and human epithelial lung cells (**A**,**C**), whereas TTF-1 was prominently overexpressed in the nuclei (**B**,**D**). Scale bars, 50 µM.

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
