# Peer review of "The Apoptosis Inhibitor Protein Survivin Is a Critical Cytoprotective Resistor against Silica-Based Nanotoxicity"

_nanomaterials, 2023, doi:10.3390/nano13182546_

Round 1

Reviewer 1 Report

1.      Please explain in detail what 1010 or 1012 NP/mL means? What is the basis for selecting these concentrations?

2.      There should be clear scales in the relevant figures and the clarity should be improved.

3.      Whether the dose of silica nanoparticles in the experimental treatment takes into account the actual dose of human exposure?

4.      Revise the upper and lower scripts of letters and numbers in some words.

5.      Some references need to be updated.

6.      Delete some irrelevant references.

Pay attention to correcting grammar and spelling errors.

Author Response

Please find the response attached as PDF.

Reviewer 2 Report

The paper entitled “The apoptosis inhibitor protein Survivin is a critical cytoprotective resistor against silica-based nanotoxicity” discussed the toxicity of silica nanoparticles and suggested a way of detoxification of them. The silica nanoparticles are expected as carrier of drug delivery, although the detailed research is not enough so far. In this aspect, this research is important. However, their discussion is not based on the figures and the supporting information was not available. Further discussion is necessary.

Line 174: 2.6 and 2.7 have same subtitles.

Line 272: Please mention the preparation of aSiNP15 in experimental section.

Line 276: The letters in Figure 1 A(bottom) and 1C(bottom), 1D(bottom) are not readable.

Line 412: There is no upper panel in Figure 5B.

Line 457: A reference is necessary to refer “our previous data”.

Line 478: In Figure, pAkt and pEPK should be defined.

Line 508: Discussion in 3.5 seems to be not supported by Figure 6 and 7.

Line 529: There are no Figure 6E-G.

Line 559: Figure 7D seems not suitable to mention the author’s opinion.

Line 569: Figure 7a (ISO-HAS-1), its figure caption (H441) and mentioned points (line 553, H441) are not matched.

Line 570: There are no explanations for Figure 7E-G.

Errors are included in the manuscript. For example,

Line 64: SiO2

Line 140: SiO2, Na2O

Line 158: strange letter was used to mention wavelength of laser.

Line 159: strange letter was used to mention wavelength of laser.

Line 178: 10 µl

Line 179: 90 µl

Line 204: CO2

Line 205: 100 µg/ml

Line 251: 1.25µg/mL

Line 590: it’s

Author Response

(The authors gave the same response as above.)

Round 2

Reviewer 2 Report

Author should mention type of the figures (e.g. fluorescence microscopy, TEM, optical microscopy and so on) in figure captions.

Line 137: Authors defined NexSil20A as aSiNP, although aSiNP seemed not to appeared in section 3. aSiNP15 suddenly appeared in section 3. aSiNP= aSiNP15? If so, please unify them.

Line 410: Was the low toxicity of aSiNP125 due to small surface area?

Line 557: where is the data for wild-type cell?
